# Photosynthetic Performance, Carbohydrate Partitioning, Growth, and Yield among Cassava Genotypes under Full Irrigation and Early Drought Treatment in a Tropical Savanna Climate

**DOI:** 10.3390/plants13152049

**Published:** 2024-07-25

**Authors:** Supranee Santanoo, Passamon Ittipong, Poramate Banterng, Nimitr Vorasoot, Sanun Jogloy, Kochaphan Vongcharoen, Piyada Theerakulpisut

**Affiliations:** 1Department of Biology, Faculty of Science, Khon Kaen University, Khon Kaen 40002, Thailand; suprsa@kku.ac.th; 2Department of Agronomy, Faculty of Agriculture, Khon Kaen University, Khon Kaen 40002, Thailand; passamon.i@kkumail.com (P.I.); bporam@kku.ac.th (P.B.); nvorasoot1@gmail.com (N.V.); sjogloy@gmail.com (S.J.); 3Faculty of Science and Health Technology, Kalasin University, Kalasin 46000, Thailand; kocha_9@hotmail.com

**Keywords:** carbon partitioning, drought stress, *Manihot esculenta*, non-structural carbohydrates, photoassimilates, tuber biomass

## Abstract

In a tropical savanna climate like Thailand, cassava can be planted all year round and harvested at 8 to 12 months after planting (MAP). However, it is not clear how water limitation during the dry season without rain affects carbon assimilation, partitioning, and yield. In this field investigation, six cassava genotypes were planted in the rainy season (August 2021) under continuous irrigation (control) or subjected to drought for 60 days from 3MAP to 5MAP during the dry season (November 2021 to January 2022) with no irrigation and rainfall. After that, the plants were rewatered and continued growing until harvest at 12MAP. After 60 days of stress, there were significant reductions in the mean net photosynthesis rate (Pn), petiole, and root dry weight (DW), and slight reductions in leaf, stem, and tuber DW. The mean starch concentrations were reduced by 42% and 16% in leaves and tubers, respectively, but increased by 12% in stems. At 6MAP after 30 days of rewatering, Pn fully recovered, and stem starch was remobilized resulting in a dramatic increase in the DW of all the organs. Although the mean tuber DW of the drought plants at 6MAP was significantly lower than that of the control, it was significantly higher at 12MAP. Moreover, the mean tuber starch concentration at 12MAP of the drought plants (18.81%) was also significantly higher than that of the controls (16.46%). In the drought treatment, the high-yielding varieties, RY9, RY72, KU50, and CMR38-125-77 were similarly productive in terms of tuber DW and starch concentration while the breeding line CM523-7 produced the lowest tuber biomass and significantly lower starch content. Therefore, for cassava planted in the rainy season in the tropical savanna climate, the exposure to drought during the early growth stage was more beneficial than the continuous irrigation.

## 1. Introduction

Cassava (*Manihot esculenta* Crantz) is one of the primary carbohydrate food sources for human consumption and animal feed for more than 0.8 billion people every year in Africa, Latin America, and Asia [1]. It is also used to produce starch for industrial applications including MSG (monosodium glutamate), glucose, fructose, sorbitol, sago, citric acid, paper, textile, plywood, glue, and ethanol [2]. Moreover, cassava leaves, stems, and extracts have wound–healing, anticancer, and hypotensive properties [3]. The price of cassava tuber is often decided by industrialists based on the percentage of starch in the tubers [4]. Cassava tubers contain a high amount of starch content that usually ranges from 70 to 90% on a dry matter basis depending on growing conditions and cassava cultivars [5,6]. Cassava grows well in humid-warm climates where temperatures range from 25 to 29 °C and soil temperature at 30 °C for maximum growth and output as well as for optimal leaf temperatures of 25–35 °C for photosynthesis [7]. Low temperature causes slow growth and reduced stomatal conductance, net photosynthesis, and yields of cassava [8,9]. Precipitations between 1000 and 1500 mm with well-distributed rainfall display favorable growing conditions of cassava [10]. The yield of cassava is also dependent on soil types. Clay and sandy soils support the growth of the leaves and stem, while loamy soil promotes and boosts root development more than the leaves and stems [11].

Environmental factors strongly influence photosynthetic performance, assimilate partitioning, and ultimately cause a significant promotion of the growth and yields of plants [12]. Water availability is one of the essential factors in determining cassava physiological expression and the yield of each genotype [13]. The primary source of water for the plant comes from rainfall and irrigation. The immediate effects of drought stress on the ability of photosynthesis involves the induction of stomatal closure, which restricts CO_2_ diffusion and directly reduces the rate of net photosynthesis [14,15]. Up to 80% of photosynthetically fixed carbon is normally exported from mature leaves. The phloem plays a major role in connecting source and sink organs and supplying sugars, mainly in the form of sucrose, to sinks [16]. The export rate of carbon assimilates from source to sink generally depends upon the rate of photosynthesis and the sugar metabolism in green leaves [17]. In the phloem of source leaves, the hydrolysis of sucrose via cell wall invertase plays a critical role in converting sugars to transient starch in leaves, thus inhibiting sucrose transport to storage roots and reducing cassava root yield [18]. The distribution of resources among sinks is also a key factor in plant productivity. Drought stress causes various undesirable effects on biomass and productivity due to abnormal physiological processes like the disturbance of water relations, osmotic adjustment, membrane integrity, carbon assimilation rate, gaseous exchange, and oxidative damage [19,20]. The susceptibility of plants to drought stress varies depending on stress degree and their developmental stages [21]. For cassava, water deficit during early growth at the plant age 1–3 months after planting (MAP) causes more yield losses as this period coincides with the critical yield attributes, viz. root system development, canopy establishment, storage root initiation, multiplication, bulking, and starch accumulation [22,23]. Under various deteriorating future climates in Thailand, the crop simulation model of cassava predicted the drought-induced yield reduction in storage root ranging from −17.24 to −21.26% [24]. In the dry upland environment in Thailand, water-limited conditions during the high storage root accumulation stage of cassava caused a substantial decline in total dry weight (DW) and storage root DW depending on cassava genotypic potentials [13]. Therefore, major breeding efforts are already in place to develop cassava cultivars that can maintain relatively high productivity under drought conditions [25].

Until the early 1990s, the most popular cassava variety in Thailand was the local variety such as Rayong 1 [2]. In 1994, the Thai Government established a special program for the rapid multiplication and distribution of new varieties with high yield potential, high harvest index, high root starch content, and early harvestability. Varieties under the name of Rayong (RY) have been released from Rayong Field Crops Research Center, Department of Agriculture, while Kasetsart 50 (KU50) has been released from Kasetsart University in 1992 [26]. The most important cassava cultivar in Thailand is KU50 whose cultivation covered 57.12% of the cassava growing area (about 633,700 ha) in 2016, while the growing area for RY5, RY90, and other genotypes was 22.21% (246,400 ha), 10.74% (119,200 ha), and 9.93% (110,000 ha), respectively [27]. Cassava is an important economic agricultural product grown in 48 of the 76 provinces in Thailand with a harvested area of around 9.87 million rai or 1.57 million ha [28]. Thailand is ranked as the world’s largest exporter of cassava products, supplying around 67% of the global market, with the other 33% going to the domestic market. During 2021–2023, the annual average of cassava production was 33.26 million tonnes of raw product [28]. Cassava has a long growth cycle (12–18 months) during which the tuber roots can be harvested from 8 to 18MAP depending on cultivars and growing conditions [29,30]. In Thailand, most of the cassava-growing areas are in the tropical savanna climate zone with a growing period, from planting to storage root harvesting, of 8 to 12 months (depending on root availability) [31], which covers almost all three seasons (hot season, March to May; rainy season, June to October; and cool season, November to February) [32].

Due to the latitude of the country (tropic region, 16°N, 101°E), seasonal variations in Thailand are under the influence of the southwest monsoon coming from the Indian Ocean and the Gulf of Bengal, providing rain to the region for about six months, i.e., from May to October, while cold air comes from mainland China, resulting in a cool and dry climate from November to February [33]. The climatic condition is suitable for cassava which can be grown all year round. Approximately 23–31% of the crop is planted in May (early hot season), 10–20% in April (hot season), 11–17% in June (rainy season), and 10–20% during October (late rainy season) and November (early cool season) [32]. Due to the growing cassava-based industry, a large and continuous supply of tubers is demanded all year round. According to Phoncharoen et al. [34] the yields of different cassava genotypes grown under irrigation varied widely depending on the time of planting, i.e., KU50 had the greatest storage root DW when planted in November and December, while RY9 and CMR38-125-77 had the highest storage root DW when planted in May, October, and November. RY11 produced the highest storage root DW when planted in April and May. The optimal planting time and the selection of appropriate genotypes are important strategies to help raise the production yield of cassava in the tropical savanna climate. Cassava productivity is determined not only by heredity but also by the interaction between genetics and environments [34]. Hence, understanding the combinatorial effect of genotypes and environment, particularly drought, on carbon assimilation, partitioning, growth, and yield is crucial for cassava improvement. Therefore, the objective of this work was to investigate the photosynthetic potentials, carbohydrate partitioning, growth, and yield of six cassava genotypes in response to drought during the early growth stage. Understanding the performance of each cassava genotype is useful for better water management to obtain an optimal tuber yield and starch content. In addition, the information will be useful for choosing suitable cassava genotypes for planting in the rainy season in the tropical savanna climate and for facilitating breeding efforts for high starch-yielding.

## 2. Results

### 2.1. The Effect of Drought Stress on Photosynthetic Performance

#### 2.1.1. Chlorophyll Fluorescence

The photochemical efficiency based on the chlorophyll fluorescence parameters of the six cassava genotypes (RY9, RY72, KU50, CMR38-125-77, CMR35-91-63, and CM523-7) growing under differing water managements (control and drought) are displayed in Figure 1 and Appendix A. Chlorophyll fluorescence parameters including the maximum photochemical quantum yield of PSII (Fv/Fm) were measured in the dark, while the effective quantum yield of PSII photochemistry (ΦPSII) and electron transport rate (ETR) were determined under natural light intensity at the plant age of 3, 4, 5, 6, and 12MAP (corresponding to the first day of withholding water, 0DAS; 30 days after withholding water, 30DAS; 60 days after withholding water, 60DAS; 30 days after rewatering, 30DAR; and harvest, 12MAP). The mean Fv/Fm, ΦPSII, and ETR across genotypes did not vary significantly between the water managements. At 30DAS, the mean Fv/Fm across genotypes significantly (*p* < 0.01) decreased from 0DAS (from 0.87 to 0.83) in both control and drought conditions and remained stable at 60DAS. Upon rewatering for 30 days (30DAR/6MAP), the mean Fv/Fm significantly increased to 0.89 in both fields (Figure 1A,B). Regardless of the water management, the mean ΦPSII and ETR across the genotypes of the young plants (3MAP/0DAS and 4MAP/30DAS) were significantly (*p* < 0.01) higher than the older plants (5MAP/60DAS and 6MAP/30DAR). The mean ΦPSII and ETR across the genotypes of the 5MAP were reduced by 42% compared with those at 4MAP. At harvest (12MAP), the mean ΦPSII significantly reduced to 0.26 and 0.28 for the control and drought groups, respectively, and the mean ETR to 148 and 141 µmol e^−^ m^−2^ s^−1^ (Figure 1C–F). For genotypic comparisons, significant differences among the cassava genotypes were observed only in the ΦPSII of the control plants at 5MAP. The ΦPSII values of CMR38-125-77 (0.39), RY9 (0.37), CM523-7 (0.35), and CMR35-91-63 (0.34) were significantly (*p* < 0.05) higher than that of KU50 (0.26) (Figure 1C). At 5MAP (60DAS), the ΦPSII of CM523-7 (0.31) was slightly lower than the other genotypes (0.32 to 0.36).

#### 2.1.2. Leaf Gas Exchange

The influences of drought stress on the net photosynthesis rate (Pn), stomatal conductance (gs), transpiration rate (Tr), and water use efficiency (WUE) of six cassava genotypes growing under different water managements are displayed in Figure 2 and Appendix A. In the well-watered control plants, the mean Pn across genotypes of the 4MAP and 5MAP plants (26 µmol CO_2_ m^−2^ s^−1^) was significantly (*p* < 0.01) higher than that at 3MAP (20 µmol CO_2_ m^−2^ s^−1^) (Figure 2A). In contrast, for the drought field, the mean Pn across genotypes of the 4MAP plants (21.65 µmol CO_2_ m^−2^ s^−1^) was similar to that of the 3MAP plants (20.5 µmol CO_2_ m^−2^ s^−1^) (Figure 2B). However, at 5MAP (60 days without irrigation, 60DAS), the mean Pn significantly (*p* < 0.01) reduced to 14.75 µmol CO_2_ m^−2^ s^−1^. At 6MAP, after 30 days of rewatering (30DAR), the mean Pn of the previously stressed plants significantly (*p* < 0.01) increased to 23.43 µmol CO_2_ m^−2^ s^−1^ which was similar to that of the 6MAP control plants (22.68 µmol CO_2_ m^−2^ s^−1^). Surprisingly, at harvest (12MAP), the mean Pn of the previously stressed plants (19.38 µmol CO_2_ m^−2^ s^−1^) was significantly (*p* < 0.01) higher than that of the control plants (12.26 µmol CO_2_ m^−2^ s^−1^) (Figure 2A,B). The significant differences in the Pn values among the cassava genotypes were observed only in the control plot at the mature stage (Appendix A). The Pn values of RY9 (17.99 µmol CO_2_ m^−2^ s^−1^), KU50 (16.63 µmol CO_2_ m^−2^ s^−1^), CM523-7 (15.81 µmol CO_2_ m^−2^ s^−1^), and CMR35-91-63 (14.06 µmol m^−2^ s^−1^) were significantly (*p* < 0.01) higher than that of CMR38-125-77and RY72, which were 4.59 and 4.50 µmol m^−2^ s^−1^, respectively.

The mean gs across genotypes of the well-watered plants varied with age, being similar at 3MAP (0.21 mol H_2_O m^−2^ s^−1^) and 4MAP (0.16 mol H_2_O m^−2^ s^−1^), hugely increased at 5MAP (0.37 mol H_2_O m^−2^ s^−1^), then dropped down to 0.21 and 0.05 mol H_2_O m^−2^ s^−1^ at 6MAP and 12MAP (Figure 2C). For the droughted plants, the mean gs across genotypes significantly (*p* < 0.01) decreased from 0.23 mol H_2_O m^−2^ s^−1^ at 3MAP to 0.09 mol H_2_O m^−2^ s^−1^ at 4MAP after 30DAS (60.8% reduction) and slightly increased at 5MAP (0.12 mol H_2_O m^−2^ s^−1^) after 60DAS. The cassava plants which received no irrigation for 30 and 60 days displayed significantly (*p* < 0.01) lower mean gs values than the well-watered plants. After rewatering for 30 days, the mean gs of the stressed plants recovered to a similar level as that of the 6MAP control plants (Figure 2D). Interestingly, the mean gs across genotypes of the stressed plants at 12MAP (0.12 mol H_2_O m^−2^ s^−1^) was significantly higher than that of the 12MAP control plants (0.05 mol H_2_O m^−2^ s^−1^). The significant differences among cassava genotypes were not observed in the gs values regardless of the treatments and plant age; however, RY9 showed slightly higher mean gs values than the other genotypes in both the water management conditions (Figure 2C,D).

The changes in Tr as affected by plant age and watering regimes in the control and stressed plants (Figure 2E,F) followed a similar pattern as that of the gs values (Figure 2C,D). Cassava plants at 4MAP and 5MAP which received no irrigation for 30 and 60 days showed much lower Tr than the well-watered plants of the same age. Similar to Pn and gs, the Tr of the 12MAP stressed plants (2.35 mmol H_2_O m^−2^ s^−1^) was significantly (*p* < 0.05) higher than that of the 12MAP control (1.06 mmol H_2_O m^−2^ s^−1^) plants (Figure 2E,F). The mean WUE across genotypes of the control plants significantly (*p* < 0.01) increased from 3MAP (4.68 µmol CO_2_ mmol H_2_O^−1^) to 4MAP (9.72 µmol CO_2_ mmol H_2_O^−1^), then slightly decreased at 5 and 6MAP (Figure 2G). The mean WUE across genotypes of the 4MAP stressed plants which received no irrigation for 30 days was 16.27 µmol CO_2_ mmol H_2_O^−1^ (a 3-fold increase from that at 3MAP). However, after 30 more days without irrigation, the mean WUE dramatically reduced to a similar level as that of 3MAP (Figure 2H). Unlike Pn, gs, and Tr, rewatering did not result in an increase in WUE, and at maturity the WUE of the control (11.94 µmol CO_2_ mmol H_2_O^−1^) was higher than that of the stressed plants (8.87 µmol CO_2_ mmol H_2_O^−1^).

### 2.2. The Effect of Drought Stress on Carbohydrate Partitioning

#### 2.2.1. Reducing, Non-Reducing, and Total Sugar, and Starch Content of Cassava Leaves

To determine the effects of drought stress on carbohydrate partitioning, reducing, non-reducing, and total sugars in the leaves of six cassava genotypes were investigated at the plant ages of 3, 4, 5, 6, and 12MAP (Figure 3 and Appendix A). The pattern of changes and quantity of reducing sugars during 3MAP to 6MAP were similar and not significantly different between the well-watered and the stressed plants. However, a significant (*p* < 0.01) difference was observed at maturity, i.e., the leaves of the control plants had significantly higher mean reducing sugars across genotypes (8.55 mg g^−1^) than that of the stressed (7.38 mg g^−1^) ones (Figure 3A,B). The significant differences in reducing sugar among the cassava genotypes were observed in the well-watered 4MAP plants, i.e., RY9 (10.05 mg g^−1^) had significantly (*p* < 0.05) higher reducing sugar than KU50 (5.10 mg g^−1^) and RY72 (3.68 mg g^−1^), while that of CMR38-125-77, CMR35-91-63, and CM523-7 varied from 6.02 to 6.44 mg g^−1^. In response to drought for 30 days, only 4MAP RY72 (Figure 3B) showed significantly higher leaf reducing sugar than that of the control (Figure 3A).

The mean leaf non-reducing sugars across genotypes of the control plants varied with age, decreasing from 3MAP (15.42 mg g^−1^) to 5MAP (10.10 mg g^−1^) then significantly (*p* < 0.01) increasing to 17.49 mg g^−1^ at 6MAP (Figure 3C). For the stressed plants, the mean leaf non-reducing sugars across genotypes also decreased from 3MAP (15.51 mg g^−1^) to 4MAP (11.98 mg g^−1^) after 30 days without irrigation. However, after 60 days without irrigation, the non-reducing sugars in the 5MAP plants significantly (*p* < 0.01) increased to 16.42 mg g^−1^, and slightly increased to 18.92 mg g^−1^ 30 days after rewatering (Figure 3D). At harvest, the mean leaf non-reducing sugars of the control (16.10 mg g^−1^) were slightly higher than that in the stressed (14.56 mg g^−1^) plants. Significant differences in non-reducing sugar among the genotypes were noted only in the 6MAP stressed plants (after rewatering), i.e., CMR38-125-77 (20.58 mg g^−1^) and CMR35-91-63 (20.34 mg g^−1^) had significantly (*p* < 0.05) higher non-reducing sugar than CM523-7 (12.48 mg/g), while RY9, KU50, and RY72 had intermediate values. The pattern of changes in total sugars in the leaves of both the control and stressed plants was similar to that of non-reducing sugar (Figure 3E,F). The line CM523-7 under stress had significantly lower non-reducing and total sugar than the remaining genotypes.

Changes in the mean leaf starch content across genotypes in the 4MAP and 5MAP well-watered plants were significantly (*p* < 0.01) lower than that in the 3MAP and slightly increased at 6MAP (Figure 3G). In the stressed 4MAP plants, after 30DAS the mean leaf starch content across genotypes was reduced by 64% (from 1.71 at 3MAP to 0.61 mg g^−1^). However, after 30 more days without irrigation, the leaf starch content remained stable. Upon rewatering for 30 days, the leaf starch content increased 2.5 times to 1.53 mg g^−1^ at 6MAP (Figure 3H). At harvest, the mean leaf starch content across genotypes in the control (1.03 mg g^−1^) plants was slightly higher than that in the stressed (0.88 mg g^−1^) ones. It is noted that during the 60-day period without irrigation, the stressed plants tended to have slightly higher total and non-reducing sugar but had much lower starch (44% reduction) compared with the controls. It is noted that at harvest, RY9 tended to have higher leaf starch content than the other genotypes.

#### 2.2.2. Reducing, Non-Reducing, and Total Sugar, and Starch Content of Cassava Stem

The mean sugar contents across genotypes including reducing, non-reducing, and total sugar of the stem (Figure 4 and Appendix A) were generally much lower than those in the leaves (Figure 4). The stem reducing sugar in the control plants tended to increase with age from 3MAP to 6MAP (Figure 4A) while that of the stressed plants remained stable from 3MAP to 5MAP, but significantly (*p* < 0.05) increased after 30 days of rewatering (30DAR/6MAP) (Figure 4B). Moreover, the mean reducing sugar across genotypes of the 5MAP stressed (1.48 mg g^−1^) plants after 60 days of stress was significantly lower than that of the 5MAP control (2.18 mg g^−1^) plants. Similarly, at harvest, the mean reducing sugar across genotypes of the stressed (3.04 mg g^−1^) plants was significantly lower than that of the control (4.79 mg g^−1^) ones. At harvest, the highest reducing sugar in the stem of the control plants was recorded in CM523-7 (11.17 mg g^−1^) followed by RY9 (6.54 mg g^−1^) while the others had intermediate values (1.62–4.67 mg g^−1^). However, the reducing sugar content in CM523-7 was drastically reduced to 4.84 mg g^−1^ in the stressed plants while that in the remaining genotypes was not significantly changed. For both well-watered and stressed groups, the mean of stem non-reducing sugar across genotypes was significantly (*p* < 0.01) reduced at 4MAP compared with that at 3MAP, and then was increased at 5 and 6MAP (Figure 4C,D). During the 60-day period of water-holding, the stressed plants tended to have lower non-reducing sugar than the controls. However, at harvest, the levels of non-reducing sugar in the stem of both groups were similar at 11 mg g^−1^ which was higher than all the other growth stages. The pattern of changes in the stem total sugar with plant age was closely similar to that of the non-reducing sugar (Figure 4E,F).

The mean stem starch content across genotypes remained stable during 3MAP to 6MAP in the well-watered plants (Figure 4G). In the stressed group, water withholding for 60 days did not affect the starch contents in the stem of the 4 and 5MAP plants (Figure 4H). Notably, the content of starch in the stem of the 6MAP plants significantly (*p* < 0.01) reduced from 2.23 mg g^−1^ in the 5MAP plants to 0.78 mg g^−1^ after rewatering (Figure 4H). It is worth noting that the stem sugar content significantly increased in response to rewatering (Figure 4B,D,F) while the starch content significantly reduced (Figure 4H). Significant differences among genotypes for the starch content in the stem were noted only in the 12MAP plants. In the well-watered plants, the maximum starch content was recorded in CMR38-125-77 (7.67 mg g^−1^) followed by CMR35-91-63 (6.65 mg g^−1^) and RY72 (6.55 mg g^−1^). These genotypes had significantly (*p* < 0.01) higher stem starch content than KU50 (4.51 mg g^−1^) and CM523-7 (2.11 mg g^−1^). For the stressed group, a significant reduction in starch content in the stem was recorded in RY9 (44% reduction from the control) and CMR38-125-77 (27% reduction from the control), while the reductions in the other genotypes were not significant. It is noted that at harvest, CM523-7 tended to have higher total and reducing sugar but much lower starch content in the stem than the other genotypes.

#### 2.2.3. Reducing, Non-Reducing, and Total Sugar, and Starch Content of Cassava Tuber

Tuber sugar contents including reducing, non-reducing, and total sugar of the six cassava genotypes in each plant age are displayed in Figure 5 and Appendix A. For the well-watered plants, the mean across genotypes for the tuber reducing sugar progressively and significantly (*p* < 0.01) reduced from 3MAP (8.06 mg g^−1^) to 5MAP (1.38 mg g^−1^) then significantly increased at 6MAP (4.53 mg g^−1^) (Figure 5A). A similar pattern was noted in the stressed plants from 3MAP (8.51 mg g^−1^) to 5MAP (2.90 mg g^−1^) and then slightly increased to 3.29 mg g^−1^ at 6MAP (after 30 days of rewatering) (Figure 5B). No significant differences in tuber reducing sugar were found among genotypes at any plant age and water management conditions. In contrast to the condition in leaves and stems, for the control plants, the mean across genotypes for tuber non-reducing sugar significantly (*p* < 0.01) increased at 4MAP (14.49 mg g^−1^) compared to that at 3MAP (9.70 mg g^−1^). At 5MAP and 6MAP, the mean values significantly reduced to 9.30 mg g^−1^ (Figure 5C). For the stressed group, the mean across genotypes for tuber non-reducing sugar of the 4MAP plants (12.21 mg g^−1^) tended to increase compared to the 3MAP plants (11.04 mg g^−1^). After 60DAS, the tuber non-reducing sugar of the 5MAP plants significantly decreased to 9.08 mg g^−1^ and continued decreasing after rewatering at 30DAR (8.14 mg g^−1^) (Figure 5D). At harvest, the mean across genotypes for tuber non-reducing sugar was similar for the control and stressed plants (15 mg g^−1^). Notably, there were prominent differences among the genotypes of the control plants at harvest, i.e., the maximum tuber non-reducing sugar content was found in RY72 (25.29 mg g^−1^) followed by RY9, KU50, and CM523-7 (showing an average of 17.23 mg g^−1^); CMR35-91-63 (10.50 mg g^−1^); and CMR38-125-77 (7.76 mg g^−1^) (Figure 5C). In the stressed group, only RY72 showed a significant reduction in tuber non-reducing sugar (17.87 mg g^−1^) which was still the highest among genotypes while CMR38-125-77contained the lowest non-reducing sugar (9.98 mg g^−1^) (Figure 5D). For the well-watered plants, the total sugar contents in the tubers of 4MAP were similar to that in the 3MAP plants (Figure 5E). In contrast, in response to water shortage, total sugar in cassava tubers tended to decrease from 19.55 mg g^−1^ (at 3MAP) to 16.24 mg g^−1^ (at 4MAP) and continued decreasing to 11.99 and 11.44 mg g^−1^ at 5MAP and 6MAP, respectively (Figure 5F). Genotypic differences in tuber total sugar followed similar patterns as that for the non-reducing sugar.

In the well-watered plants, the mean tuber starch content continuously increased from 3MAP (8.65%) to 5MAP (22.96%), then remained stable at 6MAP (21.11%) (Figure 5G). In the stressed plants, the tuber starch contents also increased with age from 3MAP to 6MAP, but the rate of increase was much lower than that in the control (Figure 5H). As a result of 30 days without irrigation, the tuber starch of the 4MAP stressed plants (13.76%) was significantly (*p* < 0.05) lower than that in the 4MAP control plants (15.44%). After 60 days without irrigation, the tuber starch of the 5MAP stressed plants (19.28%) was lower (non-significant) than that in the 5MAP control plants (22.96%). Rewatering did not have a positive effect on tuber starch content. Interestingly, at harvest, the mean tuber starch across genotypes of the stressed group (18.81%; Figure 5H) was significantly (*p* < 0.05) higher than that of the control (16.46%; Figure 5G). Significant differences in tuber starch among genotypes were recorded only at harvest. For the control group, CMR38-125-77had the highest tuber starch content (21.72%) followed by RY9, RY72, and CMR35-91-63 (having an average of 17.7%); CM523-7 (12.67%); and KU50 (11.03%). For the stressed plants, tuber starch contents were highest in CMR38-125-77 (22.21%) and RY9 (21.51) followed by RY72 (18.80%), KU50 (18.70%), and CMR35-91-63 (16.78%). The line CM523-7 had a significantly lower tuber starch concentration (14.84%) than the others (Figure 5H).

### 2.3. The Effect of Early Drought Stress on Biomass

#### 2.3.1. Above-Ground Biomass

The above-ground (leaves, petioles, and stems) dry weight (DW) of the six cassava genotypes growing under different water managements were investigated at the ages of 3, 4, 5, 6, and 12MAP (Figure 6, Appendix A). The dry weight of leaves in the control and stressed groups had similar patterns of change from 3MAP to 6MAP, and the mean across genotypes at each age was also similar between the two fields. The leaf DW of the 4MAP plants significantly (*p* < 0.01) increased (approximately 2.5 times) compared to the 3MAP ones. Due to heavy leaf senescence and abscission, leaf DW at 5MAP significantly decreased to a level similar to the 3MAP stage, but significantly increased (approximately 74% increase) at 6MAP after rewatering. At 12MAP, the mean leaf DW significantly increased (2.7- and 2.3-folds in the control and stressed plants, respectively) compared with that at 6MAP, but there were no significant differences among the genotypes in both water managements (Figure 6A,B).

The change in petiole DW with age was different from that of the leaf lamina, i.e., the mean petiole DW across genotypes continuously increased from 3MAP to 6MAP, with the highest rate of increase during the period from 3MAP to 4MAP. No significant differences were found between the two fields at 3MAP and 4MAP. However, at 5MAP, the mean petiole DW across genotypes in the drought field (23.14 g plant^−1^) was significantly (*p* < 0.05) lower than that in the control plot (37.45 g plant^−1^). After 30 days of rewatering, the petiole DW significantly increased (*p* < 0.01) in both water managements. At 12MAP, the petiole DW of the control significantly decreased from that at 6MAP (2.4 times difference) but the weight in the stressed plants decreased only slightly (Figure 6C,D).

The mean stem DW across genotypes continuously increased from 3MAP to 6MAP in the two fields, but the rate of increase tended to be lower in the drought field. Moreover, at any plant age, the mean stem DW was not significantly different between the fields. Despite the lack of water, the stem DW of the 5MAP stressed plants (164.07 g plant^−1^) was significantly (*p* < 0.01) higher than that of the 4MAP (81.69 g plant^−1^) ones (Figure 6F). At 5MAP, in the control plot, CMR35-91-63 had significantly higher stem DW than RY72, and the remaining genotypes had intermediate weights slightly lower than that of CMR35-91-63. However, in the stressed plot, all the genotypes had similar stem DW indicating that the stem growth of CMR35-91-63 was more sensitive to water shortage than others. At 12MAP, the mean stem DW of the droughted plants showed a 180% increase (*p* < 0.01) from that at 6MAP while the 12MAP stem DW of the control showed only a 42% increase (non-significant) (Figure 6E,F).

#### 2.3.2. Below-Ground Biomass

The below-ground DW (root and tuber) and total plant DW of cassava at the ages of 3, 4, 5, 6, and 12MAP in the control and stressed field are displayed in Figure 7 and Appendix A. The negative effect of soil water deficit was more pronounced in the below-ground than the above-ground biomass. The mean root DW across genotypes of the well-watered plants continuously increased from 3MAP (1.68 g plant^−1^) to 6MAP (3.51 g plant^−1^) (Figure 7A). As a result of water withholding for 30 days, the mean root DW across genotypes of the 4MAP stressed plants (0.81 g plant^−1^) was significantly (*p* < 0.01) lower than that at 3MAP (2.01 g plant^−1^). However, when water was withheld for 30 more days, the mean root DW across genotypes of the 5MAP plants increased to 1.20 g plant^−1^, and hugely increased after 30 days of rewatering to the level 3.24 g plant^−1^ (Figure 7B). The negative effects of water shortage on root growth were clearly displayed, i.e., the root DW of the stressed plants at 4MAP and 5MAP were significantly lower than those of the well-watered plants. Among the genotypes of the well-watered plants, CMR38-125-77and CMR35-91-63 at 4MAP and 5MAP had much higher root DW than the others, but all the genotypes under stress had similar root DW (Figure 7A,B). At 12MAP, the mean root DW of the control plants was significantly lower than that at 6MAP (Figure 7A), while that of the 12MAP stressed plants was not significantly decreased from that of the 6MAP ones (Figure 7B). Among the genotypes, at 12MAP, RY72 had significantly greater root DW than the others (Figure 7A), while under stress both RY72 and RY9 showed remarkable root growth (Figure 7B).

The pattern of changes in mean tuber DW across genotypes from 3MAP to 6MAP of the control and stressed plants were similar, i.e., the tuber DW continuously increased. The condition of water shortage for 60 days did not hamper the growth of the tubers, i.e., the mean tuber DW of the 4MAP and 5MAP plants of the two water conditions were not significantly different. Notably, at 6MAP, the mean tuber DW across genotypes of the control (1326.8 g plant^−1^) was significantly (*p* < 0.01) higher than that of the stressed (976.8 g plant^−1^) plants. Compared with the controls, the tuber DW of the 6MAP stressed plants was significantly reduced in RY9 (−52.8%) and CMR38-125-77 (−34.2%) while non-significant reductions were recorded in KU50 (−27.15%), CMR35-91-63 (−22.39%), and CM523-7 (−20.13%). In contrast, the tuber DW of RY72 under stress showed a 12.8% increase. The mean tuber DW at 12MAP of the control and stressed plants (1732.6) was significantly higher than that of the control (1545.8 g plant^−1^) which was 16% and 77% increased from those at 6MAP (Figure 7C,D). For the stressed group, although the tuber DWs of all the genotypes were non-significantly different, the tubers of the four high-yielding genotypes (RY9, RY72, KU50, and CMR38-125-77) tended to be heavier than the two breeding lines (Figure 7D).

Similar to the tuber DW, the total plant DW of the control and stressed plants during 3MAP to 5MAP were not significantly different between the two fields (Figure 7E,F). However, a significant difference (*p* < 0.01) was recorded at 6MAP, with the control producing 1816.6 g plant^−1^ compared with 1352.0 g plant^−1^ of the stressed plants. Compared with the controls, the total plant DW of the 6MAP stressed plants was significantly reduced in RY9 (−49.3%) and CMR38-125-77 (−31.8%) while non-significant reductions were recorded in CMR35-91-63 (−27.4%), KU50 (−25.4%), and CM523-7 (−14.6%). On the other hand, the total DW of RY72 under stress showed a 9.0% increase. The total plant DW at 12MAP significantly increased from that at 6MAP (23% and 78% increase for the control and stressed plants, respectively), but no significant difference was recorded between the two water regimes (Figure 7E,F).

### 2.4. Correlations among Physiological Parameters, Biomass, and Yield of Cassava

The relationships among physiological parameters (Fv/Fm, ΦPSII, ETR, Pn, gs, Tr, and WUE), carbohydrate content (reducing, RS; non-reducing, non-RS; and total sugar; and starch in leaves, stems, and tubers), and dry biomass (leaves, petioles, stem, root, and tuber) of the six cassava genotypes (RY9, RY72, KU50, CMR38-125-77, CMR35-91-63, and CM523-7) are demonstrated by the matrix of correlation coefficient values (r) for the control (Figure 8A) and drought conditions (Figure 8B). For the control plants, tuber DW showed a significant (*p* < 0.01) positive correlation with total DW (0.98) and non-reducing (0.61) and total sugar of stem (0.68) (Figure 8A), whereas photosynthetic parameters including ETR, ΦPSII, Tr, Pn, and gs had significant (*p* < 0.01) negative correlations with tuber DW showing the r values of −0.70, −0.67, −0.42, −0.45, and −0.36, respectively. The tuber starch content of the control plants demonstrated significantly negative relationship with tuber reducing sugar (−0.53), tuber total sugar (−0.47), leaf starch (−0.39), ΦPSII (−0.42), and ETR (−0.45), but significantly positive correlations with the dry weights of petioles (0.39), tubers (0.30), and whole plants (0.29).

For the drought plants, the tuber DW displayed significantly (*p* < 0.01) positive correlations with the total DW (0.99), stem DW (0.84), leaf DW (0.57), root and petiole DW (0.42), non-reducing (0.69) and total sugar (0.67), and starch (0.48) in stem, but had significantly negative correlations with ETR (−0.68) and ΦPSII (−0.64) (Figure 8B). The tuber starch content of the drought plants showed a significantly (*p* < 0.01) negative correlation with tuber reducing sugar (−0.59), tuber total sugar (−0.45), ΦPSII (−0.52), and ETR (−0.56), but displayed significantly positive correlations with dry weight of tuber (0.46) and total DW (0.44). The WUE of drought plants showed a significantly (*p* < 0.01) negative correlation with the Tr (−0.68), gs (−0.58), total sugar (−0.43), starch (−0.38), and non-reducing sugar (−0.32) of leaf, but that of the control plants displayed significantly negative correlation with only Tr (−0.47) and gs (−0.39). Interestingly, non-reducing sugar and total sugar in the stems of the control plants (Figure 8A) displayed significantly (*p* < 0.01) negative correlations with photosynthetic parameters (Pn, gs, Tr, ΦPSII, and ETR), but those of the drought plants (Figure 8B) displayed significantly negative correlation with only chlorophyll fluorescence parameters (ΦPSII and ETR).

### 2.5. Hierarchical Cluster Analysis of Cassava Genotypes

For the clear visualization of relationships among the cassava genotypes growing in different water regimes, agglomerative hierarchical clustering analysis (HCA) was carried out and their results are shown in Figure 9. For HCA, the data distributions of photosynthesis (Fv/Fm, ΦPSII, ETR, Pn, gs, Tr, and WUE), carbohydrate content (reducing, non-reducing, and total sugar, and starch of leaf, stem, and tuber), plant biomass (dry weights of leaves, petioles, stems, roots, and tuber) at the plant age 3, 4, 5, 6, and 12MAP were used. The heatmap clearly divided cassava into two major clusters (G1 and G2) based on physiological responses, carbohydrate partitioning, and crop biomass. All six genotypes grown under the control well-watered conditions are included in G1 which also contained CM523-7 in the drought condition, while the drought plants of the remaining five genotypes were clustered in G2. Within G1, the control plants of CMR35-91-63, RY9, CM523-7, and CMR38-125-77were grouped together in the subgroup G1a while the control plants of KU50 and RY72 were grouped together with the drought plants of CM523-7 in the subgroup G1b.

## 3. Discussion

### 3.1. The Effect of Water Stress on Photosynthetic Performance, Carbohydrate Partitioning, Growth, and Yield of Cassava

#### 3.1.1. Leaf Photosynthetic Capacity

Photosynthetic potential during the canopy establishment phase is an important physiological trait determining reducing sugars, which are essential substances for sucrose synthesis for transport to and conversion to starch in storage roots [35]. The cassava plants in this study were subjected to drought stress for 60 days during an early growth phase starting from 3MAP until 5MAP coinciding with the active canopy development stage [36]. During the 60-day drought treatment period (25 November 2021 to 25 January 2022), soil moisture at 30 cm depth in the drought plot hugely reduced from 9.96% to 6.19% and 3.96% after 30 and 60 days without irrigation, respectively (Table 1). It was clearly evident that after 30 days without irrigation and rainfall, the 4MAP cassava plants in the drought plot experienced water stress, and they responded by closing the stomata resulting in a considerable reduction in the mean gs from 0.16 mol H_2_O m^−2^ s^−1^ in the control plants (Figure 2C) to 0.09 mol H_2_O m^−2^ s^−1^ (Figure 2D). Consequently, following CO_2_ limitation, the mean Pn was significantly reduced from 26.52 µmol CO_2_ m^−2^ s^−1^ (Figure 2A) to 21.65 µmol CO_2_ m^−2^ s^−1^ (Figure 2B), i.e., an 18% reduction. With an extended period of water withholding and zero rainfall, the mean Pn of the 5MAP droughted plants was further reduced to 14.75 µmol CO_2_ m^−2^ s^−1^ which was 45% that of the well-watered plants (Figure 2A,B). It was well established that cassava is better adapted to an extended period of drought than many crops [37], and the most remarkable adaptation to drought of cassava is its ability to rapidly reduce its gs to avoid water loss through transpiration [7,38]; hence, significantly lower Tr in the droughted plants were clearly evident (Figure 2E,F). The ability of cassava to efficiently maintain leaf water status under drought was previously reported [13,39]. The superior ability to maintain leaf water status effectively protected photosynthetic apparatus particularly that of PSII [40]; hence, there were no significant differences in PSII photochemical efficiency (Fv/Fm and ΦPSII) between the control and the droughted plants after 30 and 60DAS (Figure 1A–D). However, the water-conserving strategy for survival was compromised by a reduction in CO_2_ assimilation [41]. According to Flexas and Medrano [42], among the various photosynthetic sub-processes, the Calvin cycle activity (RuBP regeneration) was rapidly inhibited during mild water stress (gs values > 0.15 mol H_2_O m^−2^ s^−1^) due to CO_2_ limitation while the photochemical activity was not inhibited until the stress became more severe (at gs values 0.05–0.15 mol H_2_O m^−2^ s^−1^). It was also noted that during the period of drought stress, the ETR was not significantly different between the control and droughted plants (Figure 1E,F) while the Pn was significantly retarded in the droughted plants (Figure 2A,B). The observation that the ETR remained high while the Pn was significantly lowered in the stressed plants implied that alternative electron transport pathways, such as the cyclic pathway around PSI, malate shuttle, and Mehler reaction, were operating under the condition of limiting CO_2_ to prevent electron carriers to become over-reduced creating reactive oxygen species (ROS) and to balance the rate of NADPH and ATP production and utilization [43,44].

After 30 days of rewatering, the photosynthetic capacity of the 6MAP stressed plants rapidly recovered so that the mean Pn (23.43 µmol CO_2_ m^−2^ s^−1^; Figure 2B) reached a similar or even slightly higher level than that of the 6MAP control plants (22.68 µmol CO_2_ m^−2^ s^−1^; Figure 2A). The recovery of the photosynthetic capacity of the stressed plants occurred concomitantly with the significant increase in gs to the pre-stress level (Figure 2D). It was reported that the newly expanded leaves of the previously stressed cassava plants showed similar or even higher Pn than those of the unstressed plants to compensate for the losses during the drought period [45]. High photosynthetic recovery upon drought release was also demonstrated in grapevine in which the Pn of the drought-rehydrated plant was higher than that of the irrigated plants to support high sink demand [46] which can be seen in this study by the high rate of biomass increase in all the sink organs (leaves, petioles, stem, roots, and developing tubers) from 5MAP to 6MAP (Figure 6 and Figure 7). The opposite pattern of change in Pn was observed in the control well-watered plants, i.e., the 6MAP plants had significantly lower Pn than the 5MAP ones (Figure 2A). This could be related to the lower light interception due to shading as evidenced by the higher leaf area index (LAI) of the 6MAP well-watered plants compared to the 5MAP ones (Appendix A).

During the 6-month growth period from 6MAP (25 February 2022) to 12MAP (25 August 2022), both the control and the previously droughted cassava plants were growing through approximately 3 months of hot season (March to May) and 3 months of rainy season (June to August), with widely different temperature, humidity, and rainfall episodes (see Figure 10). Strikingly, it was noted that the mean Pn of the 12MAP previously droughted plants (19.38 µmol CO_2_ m^−2^ s^−1^) (Figure 2B) was significantly higher than that of the control (12.26 µmol CO_2_ m^−2^ s^−1^) (Figure 2A). This indicated that the post-drought stimulation of photosynthesis observed in the 6MAP plants (after 30 days of rewatering) was sustained for several months. Similar results were reported in European beech (*Fagus sylvatica*) that the plants which experienced drought for 72 days during summer (from May to end of July) fully recovered their photosynthetic activity in 20 days after rewatering to obtain the same level of Pn as that of the control plants; thereafter, the photosynthesis of the droughted plants was stimulated and showed significantly higher Pn than the controls for the following two months until the end of the vegetative growth phase [47]. The post-drought stimulation of photosynthesis occurred to counterbalance the loss of photosynthesis and growth during the drought period [48]. The high photosynthetic activity of cassava leaves was found to be related to dramatically higher activity of the Calvin cycle and sucrose/starch synthesis enzymes compared with other plants like Arabidopsis [49].

A wide variation in Pn among cassava genotypes was previously reported [36,50,51]. Among the six genotypes investigated, it was apparent that RY9 was more tolerant of drought than the remaining cultivars displaying higher Pn during the drought period (4MAP and 5MAP) and also at the 12MAP (Figure 2B). For the well-watered 12MAP plants, RY72 and CMR38-125-77showed strikingly lower Pn than the others (Figure 2A), but these cultivars showed a high level of resilience and post-drought stimulation in photosynthesis by displaying similar Pn as the others when they experienced drought during the early growth (Figure 2B). Genotypic difference in photosynthetic capacity was also observed by Santanoo et al. [52] in that RY9 showed higher mid-day Pn than RY11, KU50, and CMR38-125-77 in the rainy and cool season. A recent study comparing the photosynthetic performance of 30 cassava genotypes revealed that RY9 showed the highest Pn under 100% and 60% crop water requirement at the mature stage (330 days after planting) [53].

#### 3.1.2. Carbon Partitioning and Growth of Cassava

The content of photosynthate available for export from a source (green leaves) depends on the carbon balance of the leaf, which is determined by its rate of photosynthesis and metabolic activity [17]. The distribution of photoassimilates in cassava plants depends strongly on the growth stage, growing conditions, and varietal differences [54]. Under drought stress (30 and 60 days after water withholding), the stressed plants of 4MAP and 5MAP exhibited approximately 45% and 42% reduction, respectively, in the mean leaf starch compared with that of the well-watered plants (Figure 3G,H) following the significant reduction in the mean Pn (Figure 2A,B). In the control condition, there was no correlation between Pn and leaf starch (Figure 8A) while a positive correlation occurred under drought (Figure 8B). This indicated that the well-watered leaves efficiently exported photoassimilates but under stress, the retardation of sugar export led to the accumulation of starch in leaves [18].

The effects of drought on the concentration of starch differed in the above- and below-ground organs. In cassava leaves, the concentration of starch was significantly reduced in the 4MAP (after 30DAS) compared with that in the 3MAP (before stress) plants in both water regimes then remained unchanged at 5MAP, and the concentrations under drought (at 4MAP and 5MAP) were lower than in the control plants (Figure 3G,H and Appendix A). This observation was similar to that reported by Duque and Setter [23] in pot-grown cassava where leaf starch considerably decreased 30 days after water stress (imposed at 2MAP) but increased after 60 days. A similar pattern of changes in leaf starch during 3MAP to 5MAP was also observed by Chiewchankaset et al. [55]. The growth stage between 3MAP and 4MAP corresponded to active canopy development as evidenced by a 2.5-fold increase in leaf DW at 4MAP (Figure 6A,B). This stage also displayed highly active tuber growth (3.3- and 4.3-fold increase in tuber DW in the control and droughted plants, respectively; Figure 7C,D) and active starch accumulation (Figure 5G,H). Hence, photoassimilates from the 4MAP leaves were mostly partitioned for the growth of these active sinks and for storage in tubers; hence, relatively small amount of carbohydrates remained in the leaves (Figure 3G,H). In stems, the starch concentrations were relatively stable from 3MAP to 5MAP in both the control and stressed plants (Figure 4G,H) while stem starch reported by Duque and Setter [23] significantly increased after 30 days of water stress and then remained stable for the next 30 days. In contrast to the pattern of changes in leaf and stem, the mean concentration of starch in tubers significantly increased with age from 3MAP to 5MAP, remained stable at 6MAP, and slightly decreased (non-significant) at 12MAP (Figure 5G,H). The same pattern of changes in starch content with age was reported in three cassava genotypes (KU50, CMR38-125-77, and RY11) planted in the early rainy season and grown under irrigation [56]. There were no significant differences in the mean tuber starch concentration between the two water regimes, except at 4MAP after 30DAS. The insensitivity of starch synthesis capacity in response to drought in cassava tuber was also reported when drought was imposed on pot-grown cassava during early growth [57] and in the rainfed field-grown cassava [58]. Our results showed that the tubers of the field-grown cassava exhibited high sink strength even under drought. In this study, the partitioning index (ratio of tuber DW and total DW) increased with plant age from 3MAP to 6MAP but was similar between the well-watered and the stressed plants (Appendix A). A similar observation was reported by Duque and Setter [23] that drought applied to young cassava plants for 60 days did not affect the sink capacity of tubers resulting in a similar partitioning index of both the control and stressed plants.

After 30 days of rewatering, the concentrations of starch in the leaves of the 6MAP plants (Figure 3H) significantly increased concomitantly with the significant increase in Pn (Figure 2B). In contrast, the stem starch was significantly reduced (Figure 4H), indicating that the stored starch was remobilized to other plant parts. It was reported that cassava stems function as an intermediary carbon storage organ during drought stress [29,38]. Significantly increased sugars in the stems after rewatering (Figure 4B,D,F) could be related to higher contents of sugars in phloem tissues indicating an enhanced sugar transport. The strong positive correlations between the non-reducing and total sugar in stems and tuber DW (Figure 8A,B) reiterated the dependence of tuber growth on efficient leaf-to-root assimilate transport [35]. Changes in the sugar and starch concentrations in the tubers in response to rewatering were different from those in leaves and stems. The concentration of tuber starch at 6MAP (after 30 days of rewatering) remained similar to that at 5MAP in both water regimes. Moreover, drought release enormously enhanced tuber growth leading to a 3.11- and 2.8-fold increase in tuber DW (at 6MAP) for the well-watered and the stressed plants, respectively, compared to that at 5MAP (Figure 7C,D). The concentrations of non-reducing sugars (mainly sucrose) in the tubers at 6MAP slightly decreased compared to those at 5MAP (Figure 5D) while the starch concentration remained stable (Figure 5H). This indicated an active conversion of imported sugars to storage starch relating to high AGPase activity and low sucrose concentration in response to rewatering [49].

During the 6-month growing period from 6MAP (25 February 2022) to 12MAP (25 August 2022), the cassava plants in both water regimes were kept well-watered, but they were exposed to high temperature and low humidity during the hot season (March to May), and high precipitation during the final growth stage during June to August (Figure 10). Despite the long period of growth with continuous irrigation, the mean total plant DW (Figure 7E) and tuber DW (Figure 7C) of the control plants at 12MAP showed only 23% and 17% increase compared to those at the 6MAP. Surprisingly, the percentage increase in total and tuber DW (from 6MAP to 12MAP) in the previously droughted plants was higher at 74% (Figure 7F) and 62% (Figure 7D), respectively. The small increase in tuber DW despite the long 6-month period of growth was presumably due to the retarded growth rate during the hot season [34] and the remobilization of starch from the tuber to support vigorous vegetative growth in the rainy season during June to August [56,59]. The observation that plants exposed to drought stress during the early growth stage performed better than the fully irrigated plants can be explained in light of the stress priming and stress memory which convey long-term acclimation after a drought stress event [48]. Several reports demonstrated that drought priming could enhance plant tolerance to subsequent drought via the functions of physiological and molecular stress memory imprints including stress-responsive osmolytes, protective proteins, transcriptional memory, and epigenetic modification through DNA methylation, histone modification, and chromatin remodeling [60,61]. Furthermore, drought priming could also enhance plant tolerance to subsequent abiotic stress such as heat stress [62]. It is possible that drought priming during 3MAP to 5MAP (November 2021 to January 2022) installed the plants with stress memory machinery which allowed the plants to better tolerate high temperature and low moisture stress during the subsequent hot season (March to May 2022) resulting in better growth during the later growth stage until harvest in August 2022. Moreover, the tubers of the 12MAP droughted plants displayed a mean starch concentration of 18.81% (Figure 5H) which was significantly higher than that of the continuously well-watered plants (16.46%) (Figure 5G). Starch yield depends both on the tuber weight and starch concentration. Therefore, for industrial consideration, the early drought treatment in this study was a more beneficial practice for farmers than continuous irrigation. The results from our study supported earlier reports that cassava exposed to early season drought during vegetative growth suffered lower tuber yield loss than when exposed later at the tuber bulking and enlargement stage [11,29]. Similar to our results, Janket et al. [63] observed that rainfed cassava (cultivar RY9) planted in the early rainy season (June) which, similar to this study, were exposed to drought during November to February produced non-significant differences in tuber DW at 12MAP as compared to the well-irrigated plants but contained a higher concentration of starch.

Genotypic differences in carbon allocation during drought, after rewatering, and finally at harvest were evident in this study. The clustering in Figure 9 indicated that most physiological parameters of all the cassava genotypes (except CM523-7) were affected by drought; therefore, genotypes in the control and stressed treatments were clearly separated. The breeding line CM523-7 tended to be least affected by drought and displayed similar levels of most physiological parameters under the control and drought conditions; therefore, the control and stressed CM523-7 were clustered in the same group (G1). Comparatively, the genotype CM523-7 tended to have lower starch concentrations in stems (Figure 4G,H) and tubers (Figure 5G,H) than the other cultivars at harvest in both water regimes. It also had relatively low tuber biomass at 12MAP under both water conditions (Figure 7C,D). However, the stressed plants of CM523-7 tended to have relatively higher stem DW at 5MAP, 6MAP, and 12MAP (Figure 6F) and especially high petiole biomass (Figure 6D). Therefore, compared to other genotypes, CM523-7 tended to allocate carbon preferentially to stems and petioles (see Appendix A) during stress and after drought release, resulting in relatively low tuber biomass at 12MAP (Figure 7D). Similar to CM523-7, the breeding line CMR35-91-63 tended to allocate more carbon to the stem (Figure 6F). Therefore, the two genotypes with relatively low tuber biomass under stress (CM523-7 and CMR35-91-63) allocated relatively more carbon to stems than the four genotypes with higher tuber biomass (RY9, RY72, KU50, and CMR38-125-77) (Figure 6F). A similar scenario was reported by Chiewchankaset et al. [55] that the modern variety KU50 (high starch and high tuber yield) was more efficient at shoot-to-root carbon partitioning while the local low-yielding variety Hanatee (low starch and low tuber yield) inefficiently allocated photoassimilates to stems rather than storage roots.

In comparison to the controls, the exposure to early drought resulted in higher total (Figure 7F) and tuber (Figure 7D) DW at 12MAP for all the genotypes (except RY72). The four genotypes (RY9, RY72, KU50, and CMR38-125-77) with relatively high tuber DW (compared with CM523-7 and CMR35-91-63) had similar tuber biomass (ranging from 1760 g plant^−1^ for RY9 to 1813 g plant^−1^ for CMR38-125-77) when subjected to early drought stress (Figure 7D). The cultivars RY9, RY72, and KU50 are recommended modern varieties widely grown commercially for a few decades while CMR38-125-77 is an advanced breeding line. These four genotypes also tended to have a higher harvest index (0.71–0.76) than CM523-7 (0.67) and CMR35-91-63 (0.68) (Appendix A). When grown under continuous irrigation, CMR38-125-77produced slightly higher tuber biomass than KU50 and RY9 which coincided with an earlier report where these three genotypes were planted in the early rainy season under full irrigation [30]. Moreover, with the observation of six planting dates (two planting dates each for the hot, rainy, and cool seasons), Phoncharoen et al. [34] concluded that CMR38-125-77was a good genotype to grow under irrigation (compared with RY9 and KU50) in terms of total and storage root DW for most planting dates. In addition, these four high-yielding genotypes had higher starch concentrations (18.70%, 18.80%, 21.51%, and 22.21% for KU50, RY72, RY9, and CMR38-125-77, respectively) than CM523-7 (14.84%) and CMR35-91-63 (16.78%). This study confirmed the results from the previous study that CMR38-125-77was the preferable genotype for planting in the early rainy season for higher starch content and starch yield [56]. In this study, although RY9 produced relatively low tuber biomass (compared with RY72, KU50, and CMR38-125-77) when fully irrigated (Figure 7C), it displayed the highest percentage increase (+28%) when exposed to early drought compared to KU50 (+16%) and CMR-38-125-77 (+6%) while RY72 did not benefit from the early drought treatment. It was noted that these four high-yielding varieties allocated relatively lower amounts of carbon to stems showing ratios of the stem to total DW ranging from 0.195 to 0.249 compared to CM523-7 (0.277) and CMR35-91-63 (0.267) (Appendix A).

## 4. Materials and Methods

### 4.1. Study Site and Environmental Conditions

The experimental field was at the field crop research station, Faculty of Agriculture, Khon Kaen University, Northeast Thailand (16°28′29.7″ N, 102°48′37.3″ E, altitude 195 m above sea level). The climatic condition in the area is a tropical savanna climate (Aw), which is most commonly found in Africa, South America, and Asia [64]. The weather conditions in the dry season can become severe, with some months having no rainfall, and drought conditions often prevail during the course of the year. For Thailand, seasonal variation is under the influence of monsoon winds, and can generally be divided into 3 seasons as follows: hot season or pre-monsoon from March to May, rainy season or southwest monsoon season from June to October, and cool season or northeast monsoon from November to February. The meteorological conditions in the field site including light intensity, temperature, and relative humidity were recorded every 30 min from the date of planting until the plants were 12 months old (August 2021 to August 2022) using an automatic recording weather station (Watchdog 2000, Spectrum Technologies Inc., Lincoln, NE, USA). The monthly mean of light intensity, air temperature, relative humidity (RH), total rainfall, and number of rainy days in the experimental field are displayed in Figure 10 and Appendix A. The light intensity means of the rainy, cool, and hot seasons were 437.8, 406.6, and 454.2 µmol (photon) m^−2^ s^−1^, respectively (Figure 10A). Air temperature means were 27.8, 24.7, and 28.1 °C, with the mean minimum of 24.1, 18.7, and 23.3 °C, and mean maximum of 32.9, 31.7, and 34.0 °C in the rainy, cool, and hot seasons, respectively (Figure 10B), while the RH means were 82.2%, 62.2%, and 70.1%, respectively (Figure 10C). Total seasonal rainfall and total number of rainy days were 1070.2 mm and 102 days, while those during the cool and hot seasons were 35.8 mm and 10 days, and 250.1 mm and 36 days, respectively. There were no rainfall in November and December 2021.

The soil in the study site is a Yasothon series (Yt: fine-loamy; siliceous, isohypothermic, Oxic Paleustults), which is distributed widely in northeast Thailand and also represents the recommended soil texture for cassava. Soil samples (0–30 and 30–60 cm depths) were taken to determine the soil texture, pH, and nutrient content. The analysis of soil texture was performed using a hydrometer, while organic matter followed Walkley–Black method [65]. The determination of soil pH, total nitrogen (Kjeldahl), available phosphorus (Bray II), exchangeable potassium (1 M ammonium acetate extraction at pH 7), electrical conductivity (EC, 1:5 H_2_O), and cation exchange capacity (CEC, 1 M ammonium acetate extraction at pH 7) was described in Baker and Norman [66]. The physical and chemical properties of the soil in the experimental field at pre-planting are displayed in Table 1. The soil texture was sandy loam, which consists of sand (>70%), silt (16.97–17.99%), and clay (7.02–12.06%). Chemical properties at soil depths 0–60 cm included available phosphorus (P) (277.5–364.5 mg kg^−1^), exchangeable potassium (K) (21.31–54.71 mg kg^−1^), total nitrogen (0.02 to 0.03%), organic matter (0.29 to 0.43%), and cation exchange capacity (3.33 to 3.53 cmol kg^−1^). The soil moisture of the well-irrigated and non-irrigated plots at the crop ages of 3, 4, 5, 6, and 12 months after planting (MAP) was measured using the gravimetric method at the soil depths of 0–30 and 30–60 cm. The field capacity and permanent wilting point were estimated using a pressure plate instrument [67]. At 3MAP (0 days after stress, 0DAS), the soil moistures were non-significantly different between the control (9.81% and 10.11% for soil depth 0–30 and 30–60 cm, respectively) and the drought (9.96% and 9.9 5%) (Table 1). At 4MAP (30DAS) and 5MAP (60DAS), soil moisture in the drought plot was much reduced compared with the control plot. The soil moisture at the 4MAP and 5MAP was 5.25% and 7.38%, respectively, at the soil depth 0–30 cm, and 1.08% and 3.58% at the soil depth 30–60 cm, respectively. At 6MAP (30 days after rewatering, 30DAR), the soil moisture of the drought plot increased to the same level as the control. At harvest (12MAP), a significant difference in soil moisture between the control and drought plots was not observed (Table 1). 

### 4.2. Plant Materials

Six cassava genotypes contrasting in crop biomass and yield were used for this study. The three commercial cassava genotypes including Rayong 9 (RY9), Rayong 72 (RY72), and Kasetsart 50 (KU50) and three breeding lines (CMR38-125-77, CMR35-91-63, and CM523-7) were evaluated under different water managements. KU50 (branching type) was recommended for the starch industry in Thailand, having high crop yield and biomass, and is grown widely across Thailand and Southeast Asia [68,69], while RY9 (non-branching type) was released in Thailand in 2005 with high yield and crop biomass [30,53]. CMR38-125-77 (branching type) was reported as having a high yield with moderate crop biomass [53], while CM523-7 (branching type) had a moderate yield with high crop biomass. RY72 and CMR35-91-63 (branching type) were reported as showing moderate yield and crop biomass [53].

Cassava cultivation started on 25 August 2021 by planting stem cuttings (20 cm in length) from 10-month-old plants. The cassava stakes of six different cassava genotypes were planted vertically on the soil ridges (2/3 of the length was buried) with plant spacing of 1 m × 1 m. The experimental design was a spilt plot in a randomized complete block design (RCBD) with four replications. The main plot of the experiment was different water managements, while the sub-plot was cassava genotypes. Each sub-plot contained 56 plants (7 plants × 8 rows) in each replication.

Land preparation was carried out following the normal procedures for the experimental fields of cassava [70]. For water management, an overhead mini sprinkler irrigation system was installed in the field to supply water to the cassava plants from planting until harvest (12 months after planting, MAP). For the control plot, irrigation was applied throughout the growing season to maintain the soil moisture content to a depth of 60 cm at field capacity level, while the drought plot was non-irrigated for 60 days at an early growth stage (from 3 to 5MAP) during the dry season (25 November 2021 to 25 January 2022). The fertilizer was applied at 1 and 2MAP based on soil analysis and the nutrient requirements for cassava [71].

### 4.3. Photosynthetic Performance

For comparing photosynthesis capacity among the six cassava genotypes including Rayong 9 (RY9), Rayong 72 (RY72), Kasetsart 50 (KU50), CMR38-125-77, CMR35-91-63, and CM523-7 growing under different water managements, photosynthesis measurements were performed on three plants/replications (*n* = 12) at five cassava plant ages (3-, 4-, 5-, 6-, and 12-month-old plants). Chlorophyll fluorescence parameters including the maximum photochemical quantum yield of PSII (Fv/Fm), the effective quantum yield of PSII photochemistry (ΦPSII), and electron transport rate (ETR) were measured on the central lobe of the first fully expanded leaf (the 5th or 6th leaf position from top) of each plant. Fv/Fm values were measured at 05:00 a.m., while ΦPSII and ETR under natural light conditions during 08:30 to 11:00 a.m. using a Mini PAM-II Photosynthesis Yield Analyzer (Heinz Walz GmbH, Effeltrich, Germany).

Leaf gas exchange parameters including net photosynthesis (Pn), stomatal conductance (gs), transpiration rate (Tr), and water use efficiency (WUE) were measured on the same leaves that chlorophyll fluorescence was measured using an infrared gas analyzer (IRGA) model Li-cor 6400xt with a LED light source using standard 2 cm × 3 cm leaf chamber (Li-Cor Inc., Lincoln, NE, USA). The measurement conditions were controlled as follows: light intensity at 1500 µmol (photon) m^−2^ s^−1^, CO_2_ concentration at 400 µmol mol^−1^, and temperature at 30 ± 2 °C.

### 4.4. Reducing Sugar, Non-Reducing Sugar, Total Sugar, and Starch

The first fully expanded leaves, stems, and tubers of the cassava plants at the age of 3, 4, 5, 6, and 12 months after planting were freshly harvested from one plant/replication, cut into 0.1 g pieces, and stored at −20 °C for the analysis of sugars and starch. The extraction for the determination of the sugar content of cassava was carried out by the method modified from [72]. The samples were finely ground and the samples (0.1 g) were extracted with 80% ethanol. The extracts were centrifuged at 12,000 rpm for 10 min at room temperature and the supernatants were collected. The extracts were kept at −20 °C until further use to determine the total soluble and reducing sugar content.

The estimation of reducing sugar was carried out using the 3,5-dinitrosalicylic acid (DNS) method following the previous report [73] with some improvements. Briefly, the extract (300 µL) was mixed with 300 µL DNS reagent (composed of 0.1 mM 3,5-dinitrosalicylic acid, 2.5 mM NaOH, and 30% potassium sodium tartrate), and 600 µL of distilled water. The mixture was heated in a boiling water bath for 8 min and then cooled down to room temperature. The absorbance was measured at 540 nm and the reducing sugar content (mg g^−1^ FW) was calculated based on a standard curve of glucose.

Total soluble sugar was estimated using the Anthrone reagent following Luo and Huang [72] with slight modifications. The sample (10 µL of the extract dilute with 90 µL distilled water) was mixed with 600 µL of freshly prepared Anthrone reagent (0.5 mM Anthrone in 70% sulfuric acid), and the tubes were boiled for 12 min. The reaction was terminated by quick cooling on ice. The absorbance was measured at 620 nm. The total soluble sugar (mg g^−1^ FW) was quantified using glucose as a standard. The non-reducing sugar was estimated from the differences between the total and reducing sugar [74].

Starch content was conducted by a modification of the method of Luo and Huang [72]. The residual materials that remained after the ethanol extraction of sugar were rinsed with 500 µL distilled water in a 1.5 mL microtube and 650 µL of cold 52% perchloric acid was added, the reactions were mixed by vortexing and then left standing at room temperature for 30 min. The samples were centrifuged at 13,000 rpm for 10 min and the supernatants were collected. The supernatant was diluted with 100 µL distilled water, mixed with 600 µL of freshly prepared Anthrone reagent, and the mixture was boiled for 12 min. The reaction was terminated by quick cooling on ice. The absorbance was measured at 620 nm. The sugars (mg g^−1^ FW) were quantified using glucose as a standard, and starch content was calculated using stock glucose 1 mg mL^−1^ [75].

### 4.5. Crop Biomass and Yields

For growth and yield measurement, four plants of each genotype (one plant/replication) were harvested at 3, 4, 5, and 6 months after planting, while 40 plants of each genotype (10 plants/replication) were used for the final harvest (12MAP). For each plant, leaves, petioles, stems, roots, and tubers were separated, dried at 80 °C for 48 h (BF 720, INDER GmbH (Headquarters), Tuttlingen, Germany) or until weight was constant, and then weighed to obtain dry weight. The cassava roots were sampled with the monolith with the size of soil volume of approximately 18,000 cm^−3^ (30 cm wide, 30 cm long, and 20 cm depth, with the plant stem located in the center).

### 4.6. Data and Statistical Analysis

The normality of data was tested using Shapiro–Wilk [76] in the Statistix version 10 software (Analytical Software, Tallahassee, FL, USA). According to data distribution, the analysis of variance according to spilt plot in RCBD was performed for assessing the significance of quantitative changes in various parameters (photosynthesis, sugar content, starch, crop biomass, and yield) in the different cassava genotypes and water managements. The main plot of the experiment was water management, while the sub-plot was cassava genotypes. For comparing cassava plant ages, the results were subjected to one-way ANOVA. Tukey’s honest significant difference test (HSD) was used for multiple comparisons of means at an alpha level of 0.05. All the graphs were taken by using the Sigmaplot Version 11.0 software (San Jose, CA, USA). The correlation among photosynthetic performance (Fv/Fm, ΦPSII, ETR, Pn, gs, Tr, and WUE), sugar content (reducing/RS, non-reducing/non-RS and total sugar of leaf, stem, and tuber), and dry biomass (leaves, petioles, stems, roots, and yields) of the six cassava genotypes was conducted in each water management. Hierarchical cluster analysis (HCA) with a heatmap was used to group cassava genotypes growing under different water managements based on photosynthetic performance, sugar and starch, biomass, and yield data. Pearson’s correlation and HCA were conducted using R version 3.4.3 [77] and Rstudio version 2023.12.1.402 [78].

## 5. Conclusions

Cassava is one of the most important cash crops for farmers in northeast Thailand due to its high growth rate and minimal requirements for agronomic resources compared with other crops. The flexibility of plant age at harvest allows cassava to be planted all year round. The results from this study demonstrated that cassava can be planted in the mid-rainy season (August) and the plants exposed to drought during the dry season (November to January) with no irrigation and zero rainfall during active canopy development and early root bulking stage produced, at 12MAP, significantly higher tuber biomass with higher starch content than those receiving continuous irrigation. Among the six genotypes tested, three commercial varieties (RY9, RY72, and KU50) and one advanced breeding line (CMR38-125-77) were equally productive while the breeding line CMR35-91-63 produced slightly lower biomass and starch content. The line CM523-7 was the least productive and contained significantly lower starch content than the others. The line CMR38-125-77, with its highest tuber dry weight and starch concentration, should be further improved to obtain a registered variety and recommended for farmers to grow in this tropical savanna climate. The length and severity of drought vary from year to year and are likely to be worsened in the future; therefore, further studies with multiple years and planting dates will be needed to effectively select the most suitable cassava genotypes for planting in each season.

## Figures and Tables

**Figure 1 plants-13-02049-f001:**
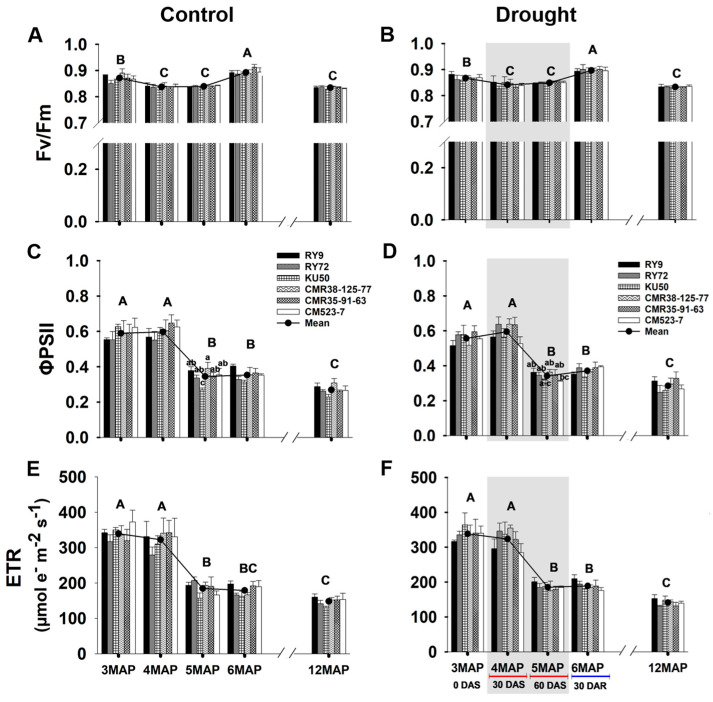
Chlorophyll fluorescence parameters of cassava genotypes Rayong 9 (RY9), Rayong 72 (RY72), Kasetsart 50 (KU50), CMR38-125-77, and CMR35-91-63 at the plant age of 3, 4, 5, 6, and 12 months after planting (MAP). The maximum photochemical quantum yield of PSII (Fv/Fm, (**A**,**B**)), effective quantum yield of PSII photochemistry (ΦPSII, (**C**,**D**)), and electron transport rate (ETR, (**E**,**F**)) were measured in cassava growing under the control ((**A**,**C**,**E**); continuous irrigation from 0MAP to 12MAP) and drought ((**B**,**D**,**F**); irrigation was withheld for 60 days in the dry season during 4MAP and 5MAP then rewatered until 12MAP) treatment. Different capital letters indicated significant (*p* < 0.05) differences among the age of plants, whereas those among genotypes are denoted with different lowercase letters in each plant age. The gray shading represents the period without irrigation and rainfall. (DAS, days without irrigation; DAR, days after rewatering).

**Figure 2 plants-13-02049-f002:**
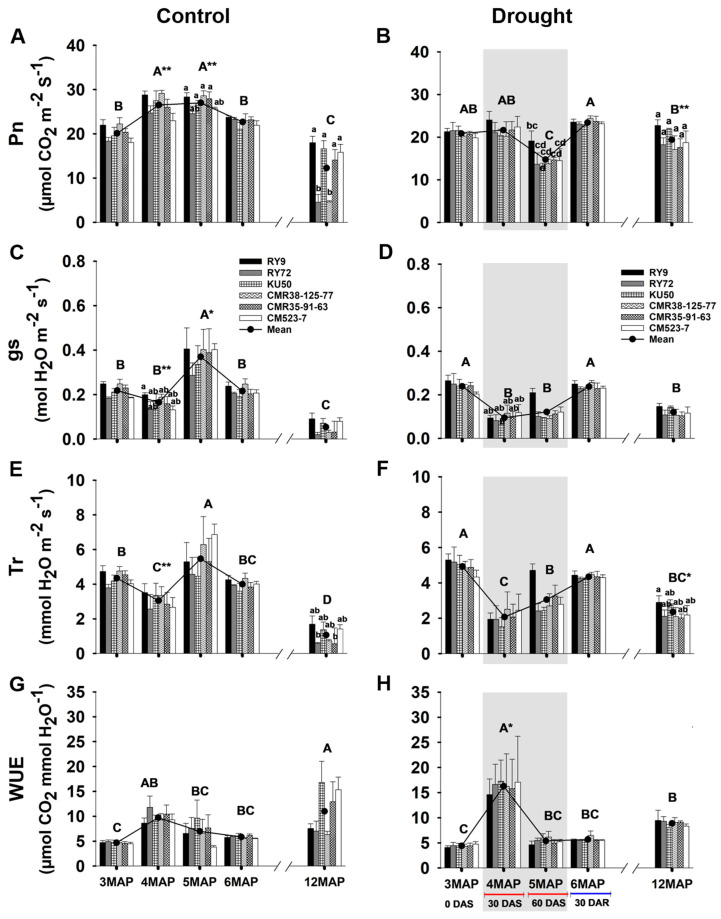
Leaf gas exchange of the cassava genotypes Rayong 9 (RY9), Rayong 72 (RY72), Kasetsart 50 (KU50), CMR38-125-77, CMR35-91-63, and CM523-7 at the plant age of 3, 4, 5, 6, and 12 months after planting (MAP). Net photosynthesis (Pn, (**A**,**B**)), stomatal conductance (gs, (**C**,**D**)), transpiration rate (Tr, (**E**,**F**)), and water use efficiency (WUE, (**G**,**H**)) were measured in the cassava growing under the control ((**A**,**C**,**E**,**G**); continuous irrigation from 0MAP to 12MAP) and drought ((**B**,**D**,**F**,**H**); irrigation was withheld for 60 days in the dry season during 4MAP and 5MAP then rewatered until 12MAP) treatment. Different capital letters indicated significant (*p* < 0.05) differences among the different plant ages, whereas those among genotypes are denoted with different lowercase letters. The significant differences (*p* < 0.05 and *p* < 0.01) between water regimes are denoted by * and **, respectively. The gray shading represents the period without irrigation and rainfall. (MAP, months after planting; DAS, days without irrigation; DAR, days after rewatering).

**Figure 3 plants-13-02049-f003:**
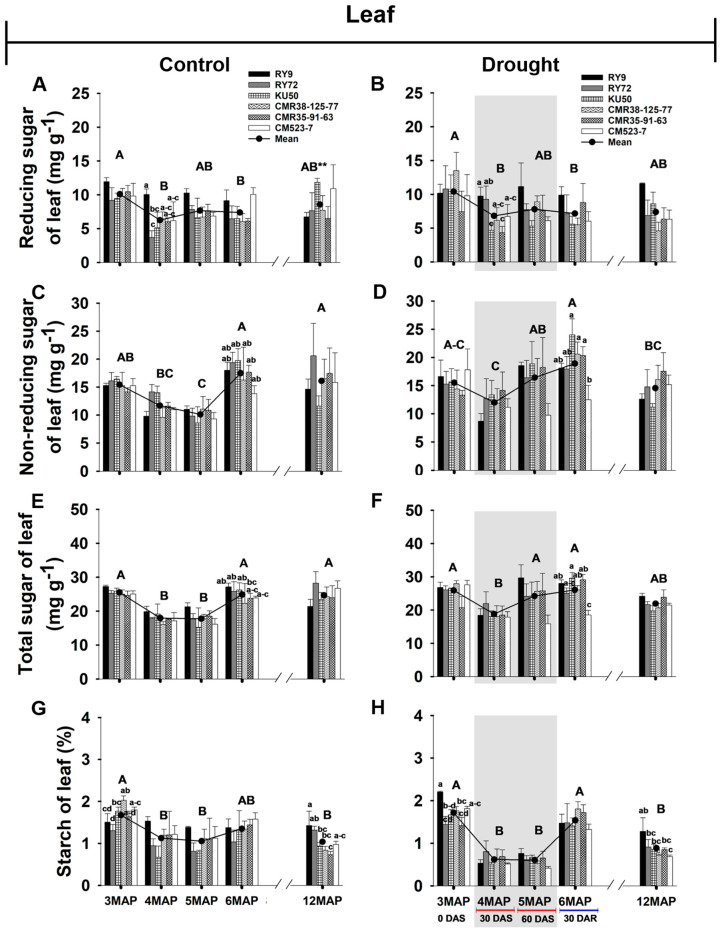
Leaf sugar (reducing, non-reducing, and total) and starch of six cassava genotypes including Rayong 9 (RY9), Rayong 72 (RY72), Kasetsart 50 (KU50), CMR38-125-77, CMR35-91-63, and CM523-7 at the plant age of 3, 4, 5, 6, and 12 months after planting. The plants were grown under the control ((**A**,**C**,**E**,**G**); continuous irrigation from 0MAP to 12MAP) and drought ((**B**,**D**,**F**,**H**); irrigation was withheld for 60 days in the dry season during 4MAP and 5MAP then rewatered until 12AMP) treatment. Different capital letters indicated significant (*p* < 0.05) differences among the different plant ages, whereas those among the genotypes are denoted with different lowercase letters. The significant differences (*p* < 0.01) between water regimes are denoted by **. The gray shading represents the period without irrigation and rainfall. (MAP, months after planting; DAS, days without irrigation; DAR, days after rewatering).

**Figure 4 plants-13-02049-f004:**
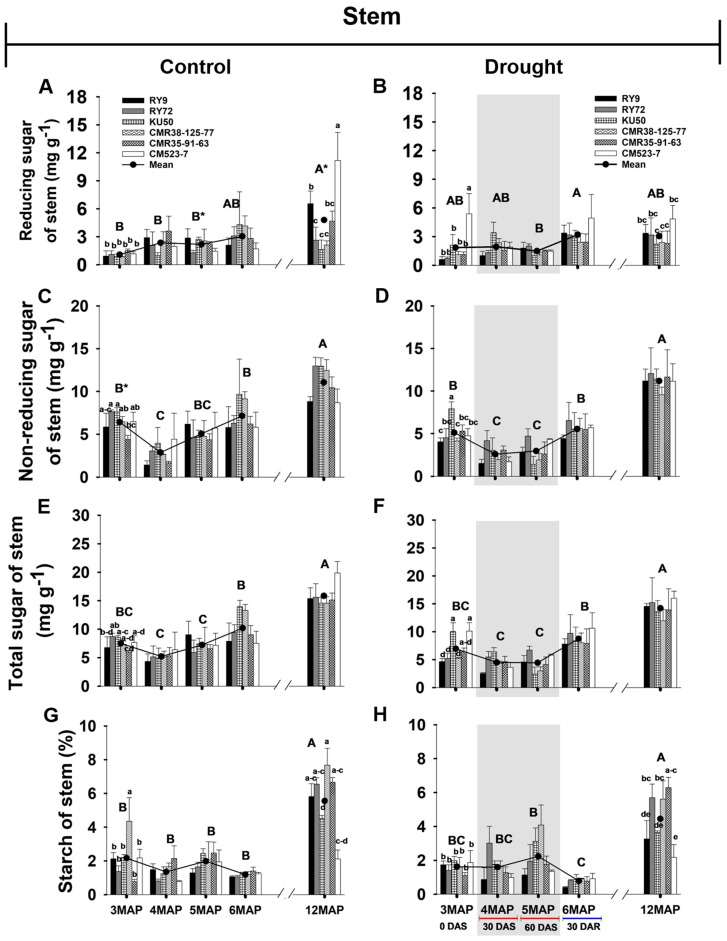
Stem sugar (reducing, non-reducing, and total) and starch of the six cassava genotypes including Rayong 9 (RY9), Rayong 72 (RY72), Kasetsart 50 (KU50), CMR38-125-77, CMR35-91-63, and CM523-7 at the plant age of 3, 4, 5, 6, and 12 months. The plants were grown under the control ((**A**,**C**,**E**,**G**); continuous irrigation from 0MAP to 12MAP) and drought ((**B**,**D**,**F**,**H**); irrigation was withheld for 60 days in the dry season during 4MAP and 5MAP then rewatered until 12MAP) treatment. Different capital letters indicated significant (*p* < 0.05) differences among the different plant ages, whereas those among genotypes are denoted with different lowercase letters. The significant differences (*p* < 0.05) between water regimes are denoted by *. The gray shading represents the period without irrigation and rainfall. (MAP, months after planting; DAS, days without irrigation; DAR, days after rewatering).

**Figure 5 plants-13-02049-f005:**
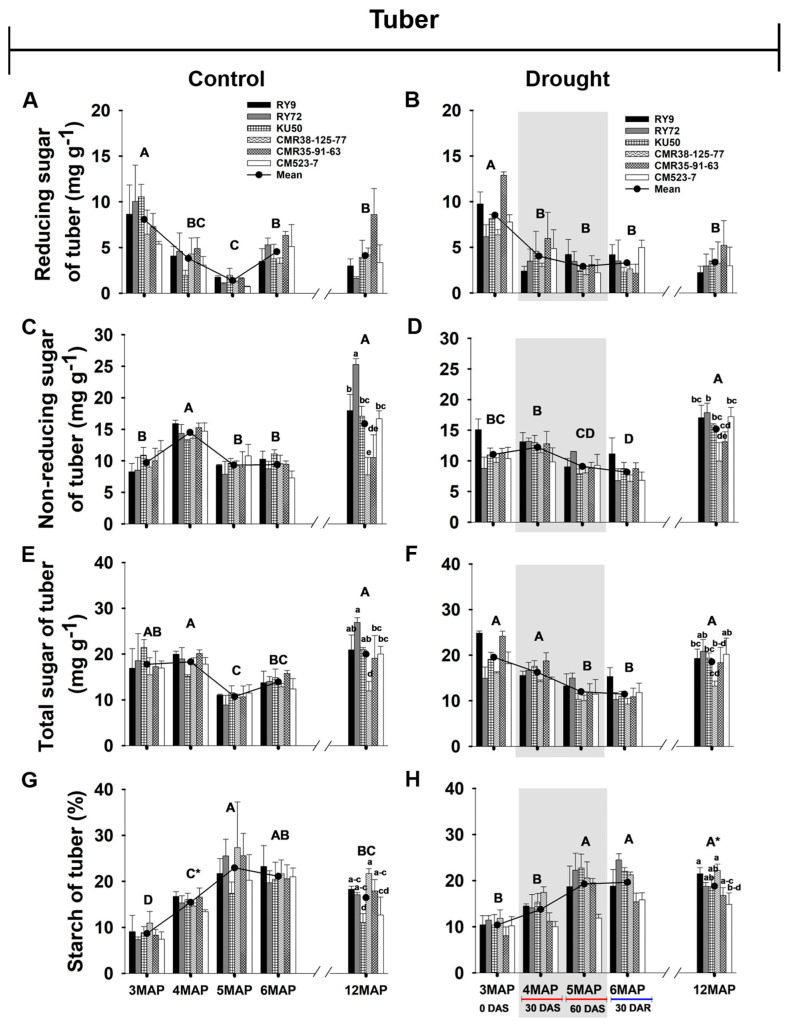
Tuber sugar (reducing, non-reducing, and total) and starch of the six cassava genotypes including Rayong 9 (RY9), Rayong 72 (RY72), Kasetsart 50 (KU50), CMR38-125-77, CMR35-91-63, and CM523-7 at the plant age of 3, 4, 5, 6, and 12 months after planting. The plants were grown under the control ((**A**,**C**,**E**,**G**); continuous irrigation from 0MAP to 12MAP) and drought ((**B**,**D**,**F**,**H**); irrigation was withheld for 60 days in the dry season during 4MAP and 5MAP then rewatered until 12MAP) treatment. Different capital letters indicated significant (*p* < 0.05) differences among the different plant ages, whereas those among genotypes are denoted with different lowercase letters. The significant differences (*p* < 0.05) between water regimes are denoted by *. The gray shading represents the period without irrigation and rainfall. (MAP, months after planting; DAS, days without irrigation; DAR, days after rewatering).

**Figure 6 plants-13-02049-f006:**
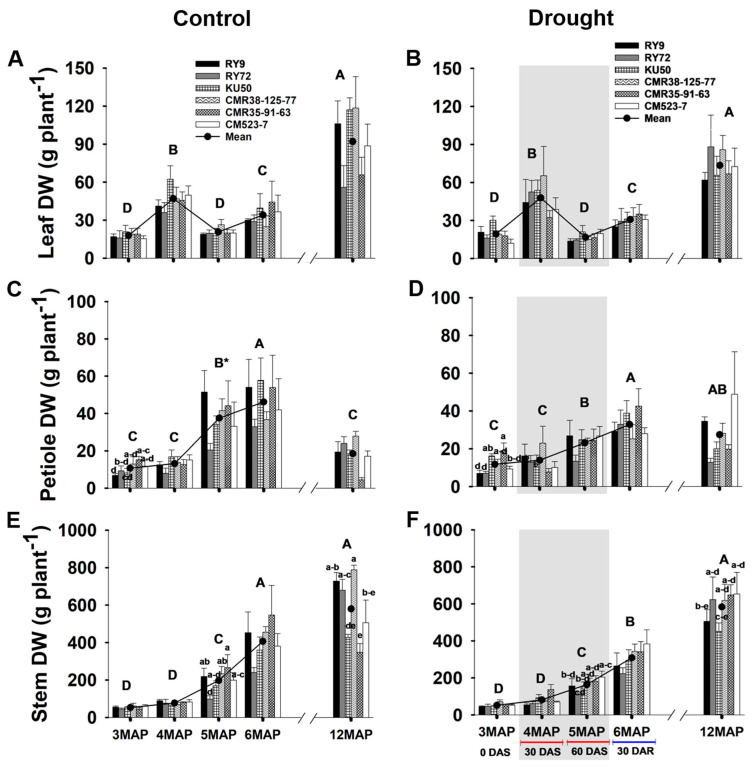
Above-ground biomass (leaf, petiole, and stem dry weight) of the six cassava genotypes including Rayong9 (RY9), Rayong72 (RY72), Kasetsart50 (KU50), CMR38-125-77, CMR35-91-63, and CM523-7 at the plant ages of 3, 4, 5, 6, and 12 months after planting. The plants were grown under the control ((**A**,**C**,**E**); continuous irrigation from 0MAP to 12MAP) and drought ((**B**,**D**,**F**); irrigation was withheld for 60 days in the dry season during 4MAP and 5MAP then rewatered until 12MAP) treatment. Each capital letter at each plant age represents the mean of the six genotypes. Different capital letters indicated significant (*p* < 0.05) differences among different plant ages, whereas those among genotypes are denoted with different lowercase letters. The significant differences (*p* < 0.05) between water regimes are donated by *. The gray shading represents the period without irrigation and rainfall. (MAP, months after planting; DAS, days without irrigation; DAR, days after rewatering).

**Figure 7 plants-13-02049-f007:**
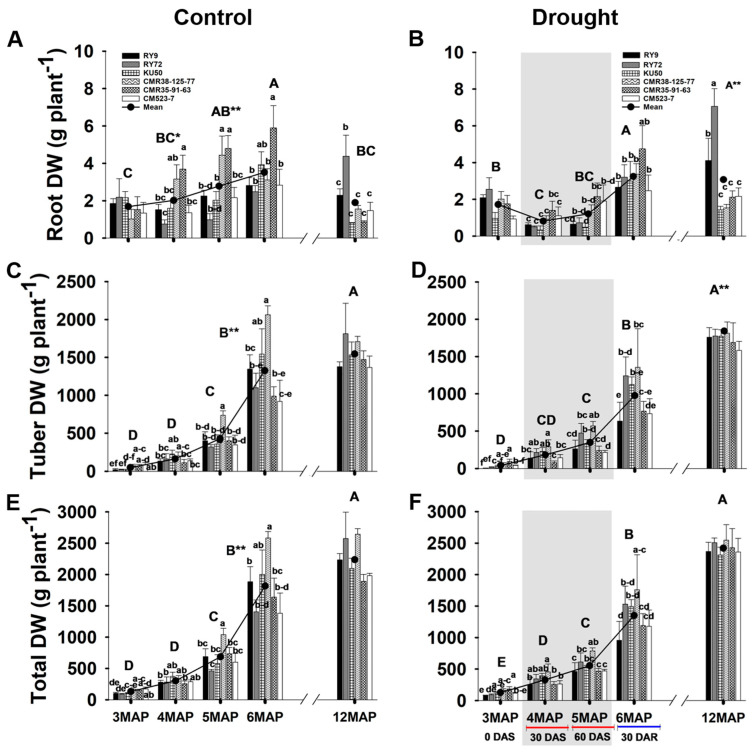
Below-ground (root and tuber) and total biomass of six cassava genotypes including Rayong9 (RY9), Rayong72 (RY72), Kasetsart50 (KU50), CMR38-125-77, CMR35-91-63, and CM523-7 at the plant ages of 3, 4, 5, 6, and 12MAP months after planting. The plants were grown under the control ((**A**,**C**,**E**); continuous irrigation from 0MAP to 12MAP) and drought ((**B**,**D**,**F**); irrigation was withheld for 60 days in the dry season during 4MAP and 5MAP then rewatered until 12MAP) treatment. Each capital letter at each plant age represents the mean of the six genotypes. Different capital letters indicated significant (*p* < 0.05) differences among different plant ages, whereas those among genotypes are denoted with different lowercase letters. The significant differences (*p* < 0.05 and *p* < 0.01) between water regimes are donated by * and **, respectively. The gray shading represents the period without irrigation and rainfall. (MAP, months after planting; DAS, days without irrigation; DAR, days after rewatering).

**Figure 8 plants-13-02049-f008:**
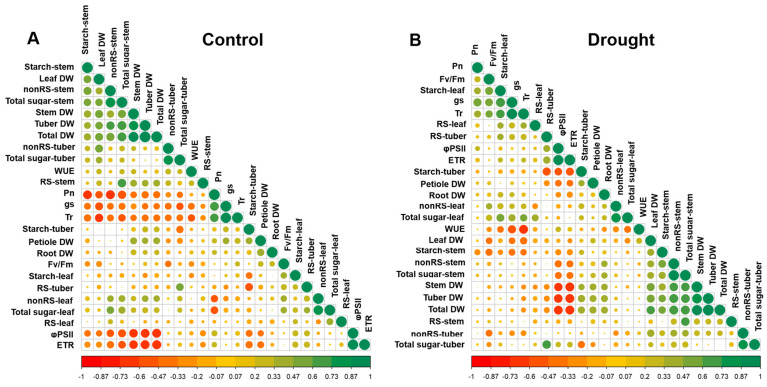
A correlation matrix of photosynthesis parameters (Fv/Fm, ΦPSII, ETR, Pn, gs, Tr, and WUE), carbohydrate contents (reducing/RS, non-reducing/non-RS, and total sugar, and starch of leaf, stem, and tuber), plant biomass (dry weights, DW of leaves, petioles, stems, and roots) and yields (tuber DW) at the plant ages 3, 4, 5, 6, and 12MAP of the six cassava genotypes (RY9, RY72, KU50, CMR38-125-77, CMR35-91-63, and CM523-7) growing under the well-watered (control) and drought (no watering for 60 days during 3MAP to 5MAP growth stage) conditions.

**Figure 9 plants-13-02049-f009:**
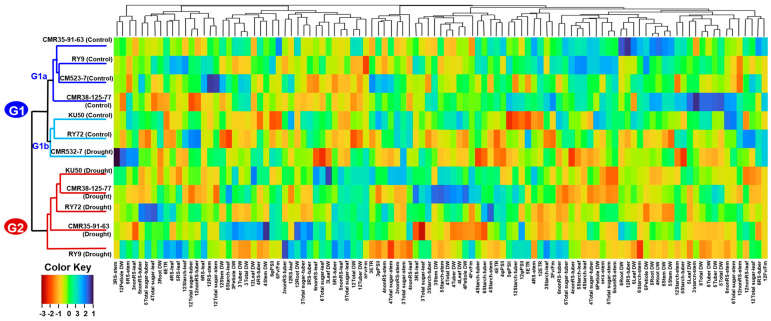
Hierarchical clustering analysis of the six cassava genotypes (RY9, RY72, KU50, CMR38-125-77, CMR35-91-63, and CM523-7) growing under different water managements. The columns correspond to the dependent variables (photosynthetic performance, sugar and starch contents, biomass, and yield), whereas the rows correspond to the different treatments (genotypes under different water managements). Low numerical values are blue, while high numerical values are red (see the scale at the left corner of the heat map; Pn = net photosynthesis rate; gs = stomatal conductance; ΦPSII = effective quantum yield of PSII photochemistry; ETR = electron transport rate; Tr = transpiration rate; Fv′/Fm′ = maximum quantum yield of PSII photochemistry in the light; WUE = water use efficiency; RS = reducing sugar; nonRS = non-reducing sugar, DW = dry weight).

**Figure 10 plants-13-02049-f010:**
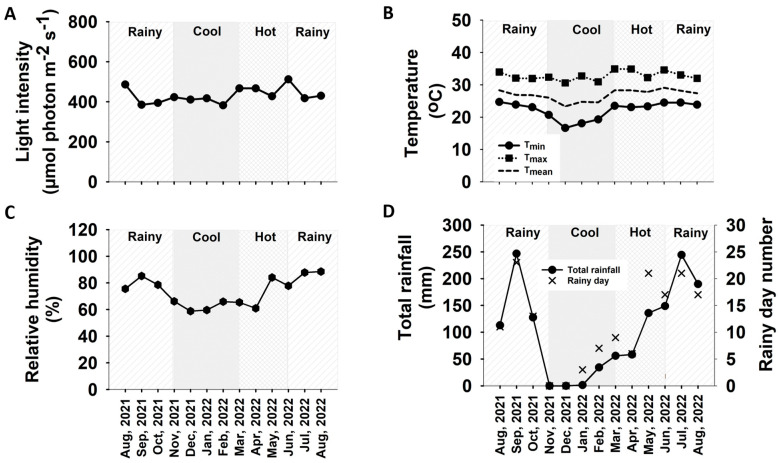
The meteorological parameters including monthly mean light intensity (**A**), air temperature (**B**) and relative humidity (**C**), total rainfall, and number of rainy days (**D**) in the cassava field during August 2021 to August 2022.

**Table 1 plants-13-02049-t001:** Physicochemical properties of soil at pre-planting in the experimental field at depths of 0–30 cm and 30–60 cm, and soil moisture (%) in the control and drought fields at different crop ages.

	Soil Physicochemical Properties			
	0–30 cm Soil Depth		30−60 cm Soil Depth	
Sand (%)	74.99		70.97	
Silt (%)	17.99		16.97	
Clay (%)	7.02		12.06	
Soil Texture	Sandy Loam		Sandy Loam	
Total N (%)	0.03		0.02	
Available P (mg kg^−1^)	364.5		277.5	
Exchangeable K (mg kg^−1^)	54.71		21.31	
OM (%)	0.43		0.29	
CEC (cmol kg^−1^)	3.33		3.59	
pH (1:1 H_2_O)	6.37		6.34	
EC (dS m^−1^)	0.03		0.02	
**Crop Age**	**Soil Moisture (%)**			
	**0−30 cm Soil Depth**		**30−60 cm Soil Depth**	
	**Control**	**Drought**	**Control**	**Drought**
3MAP (0DAS)	9.81	9.96	10.11	9.95
4MAP (30DAS)	11.44 *	6.19	9.83 *	8.73
5MAP (60DAS)	11.34 *	3.96	9.76 *	6.18
6MAP (30DAR)	12.71	11.49	11.78	10.1
12MAP	12.91	12.77	11.36	12.11

N: nitrogen; P: phosphorus; K: potassium; EC: electrical conductivity; CEC: cation exchange capacity; OM: organic matter; MAP: month after planting; DAS: day after stress; DAR: day after recovery. Significant differences (*p* < 0.05) in soil moisture between the control and drought conditions are denoted by *.

## Data Availability

All data supporting the findings of this study are available within the paper.

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
