# Peer review of "Photosynthetic Performance, Carbohydrate Partitioning, Growth, and Yield among Cassava Genotypes under Full Irrigation and Early Drought Treatment in a Tropical Savanna Climate"

_plants, 2024, doi:10.3390/plants13152049_

Round 1

Reviewer 1 Report

Comments and Suggestions for Authors

The basic ideas and results here are quite simple.  The writing borders on being too thorough in the introduction especially.  In places, the explanation goes beyond the consensus of scientific understanding.  An example of this is around line 60, where I don't think there is any consensus that stomatal closure during water deficits is "due to" ion accumulation.  Another is around line 701, where there is no real consensus regarding the feedback inhibition of photosynthesis.  I think that there are excessive references and "explanations" for a fairly simple study.

Author Response

Comments and Suggestions for Authors

The basic ideas and results here are quite simple.  The writing borders on being too thorough in the introduction especially.  In places, the explanation goes beyond the consensus of scientific understanding.  An example of this is around line 60, where I don't think there is any consensus that stomatal closure during water deficits is "due to" ion accumulation.  Another is around line 701, where there is no real consensus regarding the feedback inhibition of photosynthesis.  I think that there are excessive references and "explanations" for a fairly simple study.

Author response:- Thank you very much for your kind suggestions.

Line 60:-  ‘due to ion accumulation’ was removed and this sentence was changed to ‘The immediate effects of drought stress on ability of photosynthesis involved an in-duction of stomatal closure which restricted CO2 diffusion and directly reduced rate of net photosynthesis [14 -15].

Around Line 701:- Thank you for your comments. I agree with you that the discussion regarding the relationship between leaf sugar and photosynthesis is too long and contained excessive references, so I removed all discussions (including 5 references) in Lines 698 – 715.

Reviewer 2 Report

Comments and Suggestions for Authors

The paper is of high quality and of high importance. This is an attempt to understand biomass allocation patterns in cultivars of cassava, an important crop plant, suffering from drought stress in the initial stage of growth. The methodology is appropriate. The results are clearly presented in numerous figures and well discussed.

However some minor corrections are recommended:

Line 34, Keywords: the idea of keywords is to expand the vocabulary related to the topic of the paper so that the online search engine will help find other works on the topic of interest. I recommend to remove “cassava” included in the title and to replace it with “Manihot esculenta” and “tuber biomass”.

Line 714, Discussion: “Alves and Setter, 2004” may be replaced by [63]

Lines 920-922, Table 1: please explain clearly what differences are shown by asterix: between control and drought?

Figures 1-8 and particularly Figure 9: numerous letters and captions inside the graphs are too small and illegible. Enlargement of the letters and good resolution of the illustrations would be very helpful to the Readers.

Author Response

Response to Reviewer #2

Comments and Suggestions for Authors

The paper is of high quality and of high importance. This is an attempt to understand biomass allocation patterns in cultivars of cassava, an important crop plant, suffering from drought stress in the initial stage of growth. The methodology is appropriate. The results are clearly presented in numerous figures and well discussed.

Author response:- Thank you so much for your kind opinion of our manuscript and your constructive comments for improvements.

However some minor corrections are recommended:

Line 34, Keywords: the idea of keywords is to expand the vocabulary related to the topic of the paper so that the online search engine will help find other works on the topic of interest. I recommend to remove “cassava” included in the title and to replace it with “Manihot esculenta” and “tuber biomass”.

Author response:- The word ‘cassava’ was removed from the Keywords, and ‘Manihot esculenta’ and ‘tuber biomass’ were included.

Line 714, Discussion: “Alves and Setter, 2004” may be replaced by [63]

Author response:- This part of discussion (Line 698 - 715) were removed according to the comments of Reviewer#1, so Alves and Setter, 2004 was removed.

Lines 920-922, Table 1: please explain clearly what differences are shown by asterix: between control and drought?

Author response:- The asterisk denoted the significant difference in soil moisture between the control and drought conditions. We added this explanation into the footnote below the Table 1.

Figures 1-8 and particularly Figure 9: numerous letters and captions inside the graphs are too small and illegible. Enlargement of the letters and good resolution of the illustrations would be very helpful to the Readers.

Author response:- All the figures have been improved for better resolution and bigger letters.
